# A Comprehensive Review of Phenolic Compounds in Horticultural Plants

**DOI:** 10.3390/ijms26125767

**Published:** 2025-06-16

**Authors:** Lili Xu, Xianpu Wang

**Affiliations:** Agricultural College, Shihezi University, Shihezi 832003, China; xull_agr@shzu.edu.cn

**Keywords:** PCs, horticultural plant, metabolic pathways, stress resistance, transcription factors

## Abstract

Phenolic compounds (PCs) are key secondary metabolites in horticultural plants that are structurally categorized into flavonoids, simple phenols, stilbenes, and tannins. Synthesized via the shikimate and phenylpropanoid pathways, the metabolism of PCs is regulated by transcription factors (e.g., MYB and bZIP) and influenced by genetic backgrounds and environmental stresses (e.g., temperature and UV), thereby leading to species- or tissue-specific distribution patterns. Advanced extraction/separation techniques (e.g., ultrasonic-assisted and HPLC) have enabled systematic PC characterization. Functionally, PCs enhance plant stress resistance (abiotic/biotic) through antioxidant activity, cell wall reinforcement, and defense signaling. Their dual roles as reactive oxygen species scavengers, and signaling molecules are integral. This review synthesizes the classification, metabolic regulation, and biological functions of PCs, providing a scientific basis for improving PC content in horticultural plants with the aim of enhancing stress resilience, postharvest and storage quality, and nutritional value for sustainable agriculture.

## 1. Introduction

Phenolic compounds (PCs) are characterized by at least one aromatic ring with one or more attached hydroxyl groups, and are the most widely distributed secondary metabolites. They have a broad range of physiological roles and are mainly synthesized by higher plants, especially horticultural crops. Structurally, they encompass over 8000 molecules or derivatives, which can be divided into monomers—including flavonoids and non-flavonoids or simple phenols and stilbenes according to their molecular structures (the number of aromatic rings and their connection patterns)—and polymers, generally referring to tannins, which are divided into hydrolyzed tannins and polycondensed tannins [1]. These compounds not only endow horticultural plants with a variety of colors (such as anthocyanins determining the red, blue, and purple tones of fruits and petals) and unique flavors (such as flavonoids affecting the astringency and bitterness of fruits), but also play ecological roles in resisting environmental stress (such as salinity, extreme temperature, drought, metal toxicity, ultraviolet damage, and nutrient deficiency) and biotic stress (pathogen infection and regulating insect pollination behavior) [2,3]. At the same time, they are also an important source of antioxidant-active components in human diets [4].

Research on PCs has significantly grown with the growth of consumer demand for functional foods and the in-depth improvement of horticultural crops. On the one hand, studies on important horticultural crops such as grapes, apples, strawberries, and tomatoes have shown that the composition and content of PCs are directly related to the nutritional quality, storage characteristics, and processing adaptability of these fruits. On the other hand, integrated research based on metabolomics, transcriptomics, and gene editing technologies has revealed the dynamic regulatory mechanisms of key metabolic pathway regulatory genes, such as those relating to anthocyanins and flavonoids, as well as environmental signals, on the synthesis of PCs. However, there are still significant challenges in the current research field. The metabolic diversity of PCs among horticultural plant species has not been systematically analyzed; in particular, the identification of phenolic components in niche crops (such as characteristic flowers and wild fruit trees) is still insufficient. Furthermore, the epigenetic mechanisms by which environment–gene interactions drive the synthesis of PCs remain unclear.

This review focuses on the field of horticultural plants, systematically explains the chemical classification, metabolic pathways, extraction and separation, biological functions, factors affecting the accumulation, and regulatory network of PCs; and systematically examines the recent breakthroughs in dissecting phenolic substances within horticultural plants, leveraging multi-omics approaches and cutting-edge biotechnological tools. It critically evaluates the current challenges in this domain, while underscoring the pivotal roles of phenolic compounds in enhancing crop quality and resilience. Furthermore, the article delineates future research trajectories and translational opportunities for harnessing phenolic potential in sustainable horticulture.

## 2. Classification of PCs

Flavonoids are some of the most abundant PCs in plants, being a class of compounds with a C6-C3-C6 skeleton consisting of two aromatic rings. Based on the connection position of their three-carbon structures, flavonoids can be subdivided into different subgroups (Figure 1). Flavonoids in which the B ring is linked to positions 3 or 4 of the C rings are called isoflavones and neoflavonoids, respectively; however, substances where the B ring is connected to position 2 can be classified into multiple subgroups according to the degree of oxidation of the C ring and their functional group characteristics: flavones, flavonols, flavanones, flavanonols, flavanols or catechins, anthocyanins, and chalcones (Figure 1). They are the most diverse PCs and exist in all tissues, including leaves, stems, fruits, flowers, and roots, playing an important role in the growth and development of horticultural plants.

Additionally, simple phenols contain one or more phenolic hydroxyl groups or other substituents and have relatively simple structures; these can be divided into hydroxycinnamic acids (caffeic acid, coumaric acid, erucic acid, ferulic acid, chlorogenic acid, etc.) and hydroxybenzoic acids (protocatechuic acid, gallic acid, vanillic acid, p-hydroxybenzoic acid, etc.) [5]. Simple phenols are some of the most commonly identified and active allelopathic substances in the plant microecological environment [6]. Conversely, stilbenes are characterized by a carbon skeleton of 1,2-diphheylethylene (C6-C2-C6); an example is resveratrol, which is the most widely recognized stilbene, especially considering its role in the French paradox [7].

Tannins are polymeric compounds with a high molecular weight between 500 and over 3000 that can bind to proteins to form insoluble or soluble tannin–protein complexes. They are divided into hydrolyzable (HT) condensed tannins (CTs) and complex tannins. HTs include ellagitannins and gallotannins, which can be hydrolyzed into ellagic acid and gallic acid, respectively. CTs mainly consist of proanthocyanidins and catechin tannins and are more concentrated in plants than in HTs. Complex tannins are compounds formed by the structural units of hydrolyzable tannins and condensed tannins linked by C-C bonds.

In addition, some PCs form derivatives, such as lignins formed by alcoholization. The precursors of terpineol, sinapyl alcohol, and p-coumarol that form with phenylalanine, erucic acid, and cinnamic acid are converted into the corresponding lignin monomers through a series of enzymatic reactions [8].

## 3. Metabolic Pathways of PCs in Plants

### 3.1. Biosynthetic Pathway Centered on Phenylpropanoid Metabolism

The biosynthesis of PCs in plants begins with the shikimate pathway, through which phosphoenol pyruvic acid (PEP) and erythrose-4-phosphate (E4P) are converted into shikimic acid. This shikimic acid is further transformed into phenylalanine (Phe) and tyrosine (Tyr), serving as precursors for subsequent metabolic processes [9]. Phenylalanine acts as the direct raw material for most phenolic compounds, including flavonoids and phenolic acids, while tyrosine is involved in synthesizing lignans and stilbenes.

The phenylpropanoid metabolic pathway represents the core link in the biosynthesis of PCs. Phenylalanine is catalyzed by phenylalanine ammonia–lyase (PAL) to generate trans-cinnamic acid, which is then hydroxylated by cinnamic acid hydroxylase (C4H) to form p-coumaric acid [10]. Subsequently, p-coumaric acid combines with coenzyme A (CoA) via 4-coumarate–CoA ligase (4CL) to produce 4-coumaroyl-CoA, from which three downstream branches diverge: (1) Synthesis of phenolic acids and lignins: 4-coumaroyl-CoA is directly converted into hydroxycinnamic acids, such as caffeic acid and ferulic acid, which can further form phenolic acid esters like chlorogenic acid or lignin monomers; meanwhile, monomers such as coniferyl alcohol, derived from 4-coumaroyl-CoA, undergo oxidative coupling to form lignans (e.g., phillyrin in Forsythia). (2) Synthesis of flavonoids: 4-coumaroyl-CoA reacts with malonyl-CoA (derived from acetyl-CoA) under the action of chalcone synthase (CHS) to generate chalcone. Through cyclization reactions, chalcone gradually transforms into flavonols (e.g., quercetin), anthocyanidins (e.g., cyanidin), etc., which regulate flower color and participate in ultraviolet protection. (3) Synthesis of stilbenes: Styrene derived from tyrosine is polymerized by stilbene synthase (STS) to produce stilbenes such as resveratrol [11,12,13]. Additionally, numerous novel catalytic enzymes involved in the biosynthesis of phenolic derivatives have been identified, such as *CsSCPL4* and *CsSCPL5*, whose co-expression is likely responsible for galloylation [14].

### 3.2. Enzymatic and Non-Enzymatic Synergistic Degradation Pathways

The degradation pathways of PCs are closely associated with senescence and stress responses, categorized into two major types: enzymatic [15,16,17,18] and non-enzymatic degradation [19,20]. In enzymatic degradation, polyphenol oxidase (PPO) catalyzes the oxidation of phenolics into quinones, which further polymerize into dark-colored polymers, inducing browning in wounded plant tissues; peroxidase (POD), in the presence of H_2_O_2_, mediates the oxidative cross-linking of phenolics, playing dual roles in lignin biosynthesis and radical scavenging within the antioxidant defense system; glycosyltransferases and acyltransferases modify phenolics via glycosylation or acylation to form glycosides or esters, reducing their water solubility and sequestering them in vacuoles to indirectly regulate phenolic content; and in certain plants (e.g., legumes), symbiotic microbes secrete enzymes such as phenolic acid decarboxylases to degrade phenolics into simpler compounds. Secondly, non-enzymatic degradation mainly refers to photodegradation and chemical oxidation. Ultraviolet (UV) light cleaves conjugated double bonds in phenolics (e.g., flavonoids), leading to their fragmentation into low-molecular-weight products. As antioxidants, phenolics undergo self-oxidation through reactions with reactive oxygen species (ROS) or heavy metal ions, serving as sacrificial agents to neutralize cytotoxic molecules.

## 4. Extraction and Separation of PCs in Plants

### 4.1. Extraction of PCs

Given the diverse and extensive applications of PCs in the agriculture, food, chemical, and pharmaceutical industries, a multitude of extraction procedures for obtaining PCs from plants have emerged and continuously evolved, including traditional methods such as solvent extraction, and non-traditional methods such as ultrasonic-assisted extraction, microwave-assisted extraction, supercritical fluid extraction, and enzymatic hydrolysis.

Leveraging the solubility behavior of phytochemicals, solvent extraction remains the simplest and most foundational technique, with all modern methods essentially building upon its principles. Commonly employed solvents include organic ones such as methanol, ethanol, acetone, and water. For instance, when extracting phenolic compounds from blueberries, studies have demonstrated that organic solvents like methanol and ethanol are highly effective in extracting various PCs, including anthocyanins and phenolic acids. This efficiency can be attributed to the polar nature of these solvents, aligning well with the polarity of phenolic compounds, thereby facilitating their dissolution and extraction [21]. As reported by Pimentel-Moral et al., 200 °C and 100% ethanol represent the most favorable conditions for extracting phenolic compounds from *Hibiscus sabdariffa* [22]. Taking 50% methanol as a solvent, one study acquired 209.42 to 447.73 mg equivalent gallic acid/100 g (GAE/100 g) fresh weight PCs with an orbital shaker at a frequency of 200 rpm at room temperature. However, although this is a simple operation, this extraction method is time-consuming, and a large volume of solvent is required [23]. In another study, a green ultrasound-assisted supramolecular solvent method was developed to simultaneously determine three phenolic acids in *Prunella vulgaris* [24]. This method is also simple, with low equipment requirements, making it suitable for large-scale extraction. However, it suffers from low extraction efficiency and poor selectivity. Additionally, the extract contains a significant number of impurities, leading to complex subsequent separation and purification procedures. Moreover, the use of organic solvents may pose a risk of residue.

The ultrasonic-assisted extraction method leverages the cavitation effect to disrupt plant cell structures, facilitating the release of phenolic compounds, which also capitalizes on the mechanical effect to accelerate mass transfer and the thermal effect to enhance dissolution [25]. This method has the advantages of a short operation time, low costs, and a small volume of solvent required; however, the heat during extraction may decompose the PCs [23]. For example, when extracting PCs from rosemary, ultrasonic assistance can effectively increase the extraction amounts of phenolic components such as rosmarinic acid [26]. This method greatly shortens the extraction time, boosts the yield, and minimally damages phenolic compound structures. However, precise control of ultrasonic equipment parameters like power and frequency is crucial, as sub-optimal settings can reduce extraction efficiency and harm the bioactivity of PCs.

The microwave-assisted extraction method harnesses the rapid vibration and rotation of polar molecules within plant tissues, generating internal heat that accelerates the dissolution of phenolic compounds in cells. Simultaneously, it modifies the cell membrane’s permeability, reducing both the solvent consumption and extraction time. This technique has been widely adopted to extract PCs from various horticultural plants, such as pineapple [27], grapes [28], peach [29], and pomegranate [30]. Nevertheless, improper control of microwave intensity and duration may decompose the PCs. Additionally, the relatively high cost of the equipment and energy poses another challenge for its widespread application [23]. In addition, ultrasound- and microwave-assisted extractions can be used as efficient extraction technologies that use deionized water, 80% (*v*/*v*) ethanol in water, and ethyl acetate as solvents to extract phenolic compounds from orange peel, as they reduce extraction time and energy consumption and increase extraction yield [31].

Supercritical fluid extraction commonly employs supercritical carbon dioxide (SC-CO_2_) as the extracting agent, which combines the diffusivity of gases and the solvency of liquids. By adjusting the temperature and pressure, the dissolving capacity of SC-CO_2_ for PCs can be precisely controlled. This method has demonstrated excellent performance in extracting phenolic compounds from various horticultural plants, such as blueberries [32], rosemary [33], grapes [34], blackberries [35], and mangoes [36] with the advantage of a low operation temperature, high selectivity, inertness, and nontoxicity [23]. Nevertheless, due to the need for high-pressure equipment, the high requirement for co-solvent selection, substantial investment, and complex operation procedures, its large-scale application remains restricted.

Enzymatic hydrolysis utilizes specific enzymes (cellulase, pectinase, etc.) to destroy the structure of plant cell walls. This makes it easier to release PCs into the extraction medium [37], including tannase and β-glucosidase, which offer the highest amounts of naringenin and hesperetin in citrus pectin by-products and can improve flavanone extraction [38]. This method has mild conditions, causes little damage to the structure and activity of PCs, and can improve the extraction rate and selectivity. However, the cost of enzymes is relatively high, and the dosage of enzymes, temperature, pH value, and other conditions during the enzymatic hydrolysis process need to be strictly controlled; otherwise, this method will affect the enzymatic hydrolysis effect. In addition, during prolonged enzymatic reaction processes, impurities may also be released, which are hard to precisely control [23].

In recent years, emerging technologies such as pulsed electric field extraction, high-pressure homogenization extraction, and natural low-eutectic solvent extraction have gradually been applied for the extraction of phenols from horticultural plants. For example, the pulsed electric-assisted extraction technique significantly increased the extraction rates of PCs derived from red grape pomace [39]. Using the natural deep eutectic solvent method, 21.99 mg GAE/g of spent coffee [40] and 60.84 ± 0.48 mg gallic acid equivalent (GAE)/g dry weight (DW) PCs were recovered from Chinese nut peels (*Carya cathayensis*), with an improvement of 120% compared with the ultrasonic-assisted extraction method [41,42]. However, these new technologies still face problems such as high equipment costs, complex operations, and immature processes in practical applications, which require further research and improvement.

### 4.2. Separation of PCs

In terms of separation technology, high-performance liquid chromatography (HPLC) technology [43,44] has high resolution, sensitivity, and speed and is widely used to separate PCs from horticultural plants. Through optimizing the chromatographic conditions, phenols can be efficiently separated and purified [45], which includes the selection of appropriate chromatographic columns (such as C18 columns) and the composition and proportion of the mobile phase. This process allows for the precise separation and quantitative analysis of various structurally similar PCs, including the separation and identification of different anthocyanins in blueberries [21].

When separating PCs from Goji berries, macroporous adsorption resin based on column chromatography can better enrich and separate phenolic components such as flavonoids [46]. This method enables separation by taking advantage of the differences in the distribution coefficients of different PCs between the stationary and mobile phases. The operation is relatively simple, but the separation time is long. It may require multiple instances of elution and collection, and the separation effect on complex phenolic systems is limited [47]. Supercritical fluid chromatography uses supercritical fluid as the mobile phase and combines the advantages of gas chromatography and liquid chromatography [48]. New separation technologies, such as membrane separation [49], electrophoresis [50], and chromatography–mass spectrometry [51], have also been continuously developed and improved, providing more options for phenolic separation.

## 5. Biological Functions of PCs

### 5.1. Growth and Development

The types and contents of PCs can vary significantly between different horticultural crops. PCs such as anthocyanins accumulate in large quantities in the fruits of horticultural plants and play a crucial role in determining the final quality of the commercial product, directly determining the marketability of many fruit crops. They are found in the cell vacuole, which provides colors such as red, blue, and purple to the plant tissue per the essential structural organization of the fruit; for example, these compounds contribute to the color of red raspberries, blackberries, bilberries, blueberries, grapes, zante currants, cherries, blood oranges, red cabbage, red onions, purple sweet potatoes, red-skinned potatoes, fennels, eggplants, and radishes. The concentrations of these compounds depend on the types of tissues, growing environment, variety, harvest time, and storage conditions, providing crucial biological characteristics in the growth and development of plants.

Firstly, PCs play an indispensable role in various aspects of the plant life cycle, including seed dormancy, growth, flowering, fruit ripening, and senescence. For instance, phenolic acids and catechins in the seeds of Vitis riparia, salicylic acid (SA) in *Arabidopsis thaliana* [52,53], and gallocatechin in peas [54] participate in regulating germination. Phenolic compounds can inhibit the transport of auxin from the shoot apex to the base of the plant, thereby altering the distribution of auxin within the plant which, in turn, affects the growth and development of plant organs, influencing the elongation of roots, the formation of lateral roots, and the growth direction of stems. For example, endogenous flavonols stabilize PIN dimers to regulate auxin efflux in the same way that auxin transport inhibitor 1-naphthylphthalamic acid (NPA) influences the regulation of the polar auxin transport complex [55]. This can also affect the synthesis of plant hormones by regulating the activity of related enzymes or influence growth by regulating signal transduction; for example, as lignin does to determine the flag leaf angle in rice [56]. In one study, the highest tested tannin-to-promotor ratio strongly inhibited gibberellic acid-induced growth in pea and cucumber seedings, while also enhancing growth induced by indoleacetic acid [57]. Kim reported that ammonium regulates the amount of SA to inhibit the hypocotyl elongation of Arabidopsis [58,59].

Secondly, PCs represent a fundamental constituent of the plant cell wall. Through polymerization, they give rise to substances such as lignin, which is vital in furnishing structural support for the development of floral organs [60]. Many PCs are precursors of floral pigments, such as anthocyanins and flavonoids. These floral pigments endow flowers with rich colors, attracting pollinators, and may also play roles in the development and maturation of floral organs [61]. Additionally, they can regulate the expression of certain genes in the photoperiod pathway, influencing the plant’s response to the photoperiod and thus controlling the flowering time [62,63,64]. Chlorogenic acid is a key polyphenol in apple pulp that significantly decreases the levels of lipoxygenase, β-alactosidase, NADP-malic enzyme, the enzymatic activities of lipoxygenase and UDP–glucose pyrophosphorylase; reduces the ethylene production and respiration rate; and restrains fruit ripening and senescence [65]. Similar results have also been found in strawberries [66], grapes [67], and tomatoes [68].

The concentrations of most monomeric phenols, including chalcone, catechin, epicatechin, chlorogenic acid, ferulic acid, benzoic acid, rutin, and proanthocyanidin, decrease with the maturation of the fruit, such as walnuts [69], plums [70], and apples [71]. However, anthocyanins do not present such changes [72,73,74], and have anti-aging and antioxidative stress benefits [75]. Furthermore, polyphenols have also been shown to activate SIRT1 directly or indirectly in a variety of models, helping to regulate inflammation or cellular senescence in plants [76].

### 5.2. Antioxidant Capacity

Plants accumulate various signal substances to adapt and survive in response to environmental cues and various kinds of abiotic stresses (temperature, drought, salinity, alkalinity, and UV) (Figure 2) and pathogens/herbivores, which can cause severe damage. These stresses increase reactive oxygen species (ROS) levels, causing oxidative damage. ROS are an inevitable consequence of aerobic activities, generalized into the two major categories of oxygen-free radicals and non-radical forms. Oxygen-free radicals include superoxide (O_2_^−^), hydroxyl (OH**·**), and peroxyl (ROO**·**), and non-radical ROS include hydrogen peroxide (H_2_O_2_), singlet oxygen (^1^O_2_), and ozone (O_3_) [77]. Under normal conditions, trace amounts of ROS can serve as signaling molecules when responding to various stresses. Under severe stresses caused by ROS over-accumulation as a result of ROS-maintaining mechanism dysregulation, ROS can directly attack biological macromolecules, leading to oxidative damage to plasma membranes, nucleic acids, and proteins, and even to the oxidative obliteration of the cell [77,78]. Enhancing plant antioxidant capacity helps to improve plant stress resistance. In addition, oxidative stress activates multiple stress response mechanisms in plants, including various enzymatic antioxidants and non-enzymatic antioxidants, including ascorbic acid and glutathione, as well as PCs, such as anthocyanins and stilbenes in grapefruits [79], phenolic acids in apples [80], catechins and flavonol glycosides in black tea [81], and flavonoids in various plants [82,83]. The antioxidant capacities of various PCs are closely related to the conjugated structure of the benzene ring [84,85,86].

#### 5.2.1. Abiotic Stress

The quality and yield of many horticultural agricultural products are restricted by various environmental factors, including water stress, high salinity, extreme temperatures, mineral nutrient deficiency, metal toxicity, ultraviolet radiation, bacteria, fungi, and insect pests (Figure 2). Therefore, plants have to adapt to fluctuating environments resulting from environmental pressures, and the accumulation of phenolics in cells is a plant’s accommodating response to detrimental environmental situations [87].

Firstly, phenolic-saturated cell walls act as effective barriers in plants, restricting water movement from cells to the external environment, thus enhancing water-use efficiency, drought-survival chances, and post-drought regeneration ability [88]. Multi-omics analysis indicates that the phenylpropane pathway—particularly the flavonoid pathway—plays a crucial role in conferring salt tolerance to roses [89], alfalfa [90], strawberries [91], apple [92], and grapes [93].

Furthermore, extreme temperature stress is one of the most significant abiotic stresses confronted by plants under global climate change. Rosmarinic acid treatment induces multiple high temperature-responsive transcription factors, including HsfA2 family genes in tomatoes, potentially contributing to the induction of heat shock proteins (HSPs) to protect plants against the detrimental effects of high temperatures [94]. Research has verified that phenolic compounds can also protect plants from low-temperature stress in tomatoes [95], apples [96], grapes [97], and peaches [98]. Cold stress cooperates with SA signals to affect plant immune responses in tomatoes [99].

Pretreatment with 100 μM dopamine can promote the uptake of essential nutrients, including N, P, K, Ca, Mg, Fe, Mn, Cu, Zn, and B, to modify the pattern by which these nutrients are distributed throughout the plant [100]. Flavonoids positively influence high-soil nutrients such as Ca, Mg, Mn, and Fe in *Moringa oleifera* Lam [101]. Bali et al. found that priming tomato seeds with jasmonic acid increased the levels of secondary metabolites—substances involved in metal ligation, organic acids, and polyamines—to mitigate Pb toxicity [102].

Finally, UV-B radiation has been suggested as an additional light source during the cultivation of plants to enhance flavonoid composition, thus improving stress resistance [103], including in grapes [104], thalloid liverworts [105], zinnia profusion [106], *Salicornia rubra* [107], and *Capsicum annuum* [108].

#### 5.2.2. Biotic Stress

PCs play an important role in the process of plants coping with microbial stress. They participate in the defense responses of plants through various pathways and regulate the interactions between plants and microorganisms.

On the one hand, root-secreted PCs exhibit two functions: inhibiting detrimental microorganisms while enriching and stabilizing beneficial symbiotic microbial communities. Antibacterial properties can disrupt the cell membranes of microorganisms and interfere with their metabolism. Furthermore, PCs also reinforce the cell wall at the challenge site, accompanied by ROS driving cell wall cross-linking, antimicrobial activity, and defense signaling [109,110]. Multiple simple and complex PCs, including cajanin, medicarpin, glyceolin, rotenone, coumestrol, phaseolin, phaseolinin, and flavonoids, act as phytoalexins, phytoanticipins, and nematicides [110,111]. The polyphenol extract from flaxseed exhibits antibacterial effects against *Staphylococcus aureus*, *Escherichia coli*, *Listeria monocytogenes*, *Salmonella*, and *Pseudomonas fluorescens*. Its antibacterial mechanism mainly involves disrupting the cell membranes and cell walls of microorganisms, thus interfering with their normal life activities [112]. In addition, microorganisms directly or indirectly affect the quality and quantity of rhizosphere-localized PCs by modifying exudation patterns and microbial catabolism, influencing plant–microbe interactions [110,113]. In terms of specific symbiotic relationships, PCs secreted by plant roots can alter the rhizosphere microbial community, promoting the growth of beneficial bacteria and inhibiting pathogenic bacteria. PCs secreted by symbiotic legumes, especially flavonoids, can serve as chemoattractants for guiding the growth of rhizobial cells [110]. Acting as the signaling molecules, the host root-secreted PCs can regulate *nod* gene expression in symbionts (Rhizobium) and modify legume–rhizobial symbiosis [110]. Phenolic acids, such as p-coumaric, caffeic, and ferulic acids, serve as novel nod gene inducers in *L. japonicus*–Mesorhizobium symbiosis [114].

Certain research findings indicate that phenolic acids can facilitate the proliferation of pathogenic bacteria. The interaction between the microorganisms in rhizospheric soil and phenolic acids is the primary factor that disrupts the microflora in the rhizosphere of *Panax notoginseng* [115]. Moreover, PCs are more effective against Gram-negative bacteria, while Gram-positive bacteria are more sensitive to flavonols [116]. Flavonoids in fengdan peony demonstrate a better inhibiting effect on Gram-positive bacteria [117], and extracts from grape seeds inhibit the growth of Gram-positive and Gram-negative bacteria (GSE) [118]. Wang et al. discovered that flavonoids in *Sedum Aizoon* L. act against *Pseudomonas fragi* [119]. Meta-analyses demonstrate that phenolic induction is a common response in plant hosts exposed to feeding or colonization, with specific exceptions such as pathogenic fungi and piercing-sucking insects [120]. Tannic acid defends against the fall webworm (*Hyphantria cunea*) by inhibiting the enzyme activities of pathogenic bacteria and disrupting the cell membranes of microorganisms [121].

However, PCs can also act as signaling molecules to induce plant defense responses, activate the expression of defense genes and oxidative stress responses, and regulate plant hormone signaling pathways. SA and JA signals play important roles in shaping the outcome of plant–pathogen interactions [122]. For example, endogenous SA signaling machinery counteracts the responses elicited by either coronatine or JA in arabidopsis [123]. SA-induced redox modifications and glutathione biosynthesis play pivotal roles in suppressing the JA signaling cascade [122,124]. Ding et al. [125] observed that jasmonic acid and flavonoids increase with prolonged fall webworm feeding, suggesting their crucial role in response to pest infestation. A signaling circuit is formed by methyl-salicylate (MeSA), salicylic acid-binding protein-2 (SABP2), the transcription factor NAC2, and salicylic acid-carboxylmethyltransferase-1 (SAMT1), mediating the airborne defense (AD) of plants against aphids and viruses [126].

## 6. Factors Affecting the Accumulation of PCs

### 6.1. Genetic Factors

The composition and distribution of PCs are primarily determined by genetic factors but also vary with the maturation stage, as well as due to seasonal conditions and the physical and chemical characteristics of the soil. They predominate in plant leaves, petals, fruits, seeds, and other organs [127,128,129,130] and exhibit significant variations across different species and within distinct plant tissues [131,132]. Due to their distinct genetic backgrounds and evolutionary histories, horticultural plants exhibit significant variations in their PC contents and compositions (Table 1). Fruit trees such as grapes [133], strawberries [134], apples [135], hawthorn [82], and blueberries [136] are rich in a diverse range of phenolic compounds, including anthocyanins, flavonols, and procyanidins. Simple phenolic compounds (p-hydroxybenzoic acid, vanillic acid, ferulic acid, sinapic acid, caffeic acid, and p-coumaric acid) often exist in vegetables either in a free state or in combination with other compounds [137], such as chlorogenic acid in peppers (Solanaceae family) [138] and quercetin in onions [139]. Anthocyanins are end-products of the flavonoid pathway, generally accumulating as pigments in ornamental plants and exhibiting an extraordinary degree of divergence in different ornamental species correlated with color space values [140,141]. Meanwhile, catechins are the main polyphenol compounds in tea (*Camellia sinensis*) [142].

Moreover, within a given species, the monomer types and content distributions of various PCs vary greatly between varieties; for example, in grapes, delphinidin-derived anthocyanins represent the primary class in the peel of eight varieties. In Vitis vinifera cultivars, methyl cyanidin-derived anthocyanins typically rank as the second most abundant, whereas in American–European hybrid grapes, methyl delphinidin-derived anthocyanins dominate this secondary position [160]. Similarly, the types of anthocyanins vary between different chrysanthemum cultivars, which predominantly comprise cyanidin glycosides such as cyanidin-3-glucoside (C3g) and cyanidin-3-(3′′-malonyl)-glucoside (C3mg) [161]. Citrus fruits such as oranges and grapefruits are abundant in a variety of flavonoids, including hesperidin, naringin, nobiletin, and tangeretin [162,163,164]; however, mandarins exhibit the highest levels of both total phenolics and total flavonoids, and kumquats possess the highest PC content, of which protocatechuic acid is predominant [165]. In horticultural plant species, the total phenols, anthocyanins, and proanthocyanidin of red-skinned varieties are usually significantly higher than those of white/yellow-skinned varieties, such as potatoes [166], grapes [167], and apples [168], with fewer flavonols, such as quercetin, kaempferol, and their derivatives [167]. In addition, monomer PCs in traditional varieties are higher than those in modern improved varieties. For example, the total phenolic contents in wild strawberries, raspberries, blackberries, and apples are two to five times higher than those in cultivated plants [132,169,170]. The tannin contents of wild chickpeas are significantly higher than those of commercial varieties [171]. Chromosome-scale genomics, metabolomics, and transcriptomics analysis have revealed that *Vitis*. *adenoclada* varieties are rich in phenolic acids and flavonols, whereas flavan-3-ol and anthocyanin contents are lower than those of other varieties, such as ‘Petit Verdot’ [172].

Furthermore, marked variations exist in both the composition and concentrations of phenolic compounds across distinct tissue regions. In general, the total phenol content in a fruit peel is significantly higher in the pulp; it then decreases throughout fruit development [173,174,175]. In research on the vertical distribution of phenolic compounds in extracts from diverse plant organs of filipendula ulmaria, there were substantial disparities in both the qualitative and quantitative compositions of phenolic compounds across different tissue parts. Specifically, leaves and flowers exhibit relatively high flavonoid content, fruits possess relatively high hydroxycinnamic acid content, roots contain a relatively high amount of catechins and proanthocyanidins, and fruits have a relatively high tannin content [176].

### 6.2. External Factors

Apart from differences in species, varieties, and tissues, PC contents are also affected by non-genetic factors. Nevertheless, the majority of reference points indicate that environmental conditions and viticultural practices have a more significant impact on PC concentrations than on their relative distribution (Table 2).

Firstly, the accumulation of phenolic compounds results from modulation in the phenyl–propanoid pathway, as many of the most important protein-encoding genes of these pathways are regulated by abiotic stresses, stimulating phenolic compounds. For example, the enhanced accumulation of kaempferol and quercetin helps tomatoes to cope with drought conditions [188]; this also applies to cinnamate 4-hydroxylase and/or p-coumarate 3-hydroxylase for maize [189], anthocyanins and two flavones for *Ligularia fischeri* [190], eugenol for tea plants [191], and vanillic acid for *Cucumis sativus* [192]. The heat shock factor of MdHSFA8a is released from MdHSP90-MdHSFA8a complexes and interacts with MdRAP2.12 to activate genes for flavonoid biosynthesis, resulting in a high level of flavonoids and enhancing drought tolerance in *Malus domestica* [193]. In addition to water stress, when horticultural crops are exposed to salt stress, PCs will also significantly accumulate in the plants. For example, findings have revealed that the over-expression of NtERF4a significantly promotes the accumulation of chlorogenic acid and flavonoids in tobacco leaves, improving salt stress resistance [194]. Treatments with certain PCs, such as sinapic acid, caffeic acid, ferulic acid, and p-coumaric acid, can markedly improve salt stress tolerance in Chinese cabbage [195].

Secondly, light is a vital environmental factor that influences plant growth and development and the synthesis of PCs. Long-day or short-day conditions might activate or inhibit key genes in the phenylpropanoid metabolic pathway, affecting the accumulation of PCs. For example, anthocyanin accumulation has been reported to be light-dependent in *ZmLC*-transgenic apple leaves [196], as well as in petunia [197] and *alfalfa* [198]. Generally, appropriate light intensity impacts the efficiency of photosynthesis, which is beneficial for the accumulation of PCs in plants. In ginger (*Zingiber officinale*), for instance, there is a positive correlation between the accumulation of PCs and light intensity [199]. Light quality is one of the key elicitors that directly affect plant cell growth and the biosynthesis of phenolics. Data show that blue light is the most effective treatment for promoting the accumulation of total phenolics, and anthocyanin accumulation can be induced by white, blue, and purple light in the red callus of spine grapes. Conversely, flavonoid accumulation is favorable under mixed-wavelength light. Furthermore, proanthocyanidins accumulation is facilitated by blue and purple light [200].

Finally, the differential accumulation of phenolic compounds in response to temperature reflects an adaptive strategy formed during long-term plant evolution, which is coordinately regulated by the “species gene–tissue function–environmental signal–metabolic trade-off” framework. For instance, the leaves of one species accumulate phenolics under low temperatures to enhance freezing tolerance, whereas another species may exhibit an opposite trend due to genetic variations or different metabolic priorities [91,177]. High temperatures enhance the accumulation of condensed tannins in *Achillea millefolium* [201] and increase the total phenolic compounds in cauliflowers [202] and mediterranean aromatic plants [203], but reduce the total PC contents of tomatoes in greenhouses [179]. However, low temperatures always induce anthocyanin synthesis in plants, including cabbage [204], grapes [205], apples [206], tomatoes [207], cassavas [208], and citrus [209]. This is more pronounced with significant diurnal temperature variation, such as in wild apples [210]. However, prolonged low temperatures will inhibit fruit growth and development; for example, in strawberries, low temperatures inhibit anthocyanin accumulation and delay ripening [211] (Table 2).

## 7. Regulatory Networks of PCs

The biosynthesis and accumulation of PCs are intricately regulated by complex molecular networks, with transcription factors (TFs) occupying a central role. TFs exert precise control over the biosynthetic pathways of PCs by binding to the promoter regions of genes involved in phenolic compound synthesis, thereby activating or repressing gene expression in a context-dependent manner to regulate the accumulation profiles of PCs in plants.

In plants, a variety of proteins can bind to specific DNA sequences. They are involved in regulating the initiation of gene transcription and modulating the transcriptional level of genes of PCs; in this context, the MYB, bZIP, AP2/ERF, MADS–box, NAC, bHLH, and WRKY transcription factors have been the most extensively studied using omics technologies [209,212,213]. For example, SmMYB1 significantly promotes the accumulation of phenolic acids and anthocyanins by activating the expression of cytochrome P450-dependent monooxygenase (*CYP98A14*), chalcone isomerase (*CHI*), and anthocyanidin synthase (*ANS*) in *Salvia miltiorrhiza* [214]. Similarly, FaMYB63 directly activates *FaEGS1*, *FaEGS2* (eugenol biosynthesis), *FaCAD1* (cinnamyl alcohol dehydrogenase 1), *FaEOBII* (EMISSION OF BENZENOID II), and *FaMYB10* to induce eugenol biosynthesis during strawberry fruit development [215]. Moreover, two GOLDEN2-LIKE genes directly regulate the transcription of CsMYB5b to promote catechin accumulation in tea leaves [216]. GbMYB11 enhances the expression of *GbF3′H* and *GbFLS* to regulate flavonol biosynthesis in *Ginkgo biloba* [217]. The bZIP transcription factor MdHY5 positively regulates *MdMYB10* and promotes anthocyanin accumulation; then, the ubiquitin-26S proteasome system is employed by MdBT2 to suppress *MdbZIP44*, which promotes anthocyanin accumulation in response to ABA by enhancing the binding of MdMYB1 to promoters of downstream target genes in apples [218,219]. Knocking out one allele of *VvbZIP36* promotes anthocyanin accumulation including naringenin chalcone, naringenin, dihydroflavonols, and cyanidin-3-O-glucoside but inhibits the synthesis of stilbenes, lignans, and flavonols in grapevine [220]. The bZIP transcription factor SlAREB1 can regulate anthocyanins by controlling *SlDFR* and *SlF3′5′H* (structural genes involved in anthocyanin accumulation) expression during the low-temperature responses of tomato seedlings [207]. PC production also increases in *SmERF115*-overexpressing hairy roots but with a decrease in the RNAi hairy root lines of *SmERF115*, which can activate *SmRAS1* expression (a gene involved in PC biosynthesis) in vivo [221]. When *CsERF003,* a flavonoid activator, is overexpressed in tomato, two main flavonoids—naringeninchalcone and kaempferolrutinoside—will increase by an average of 7.99 times and 36.83 times, respectively [222]. Specifically, genes involved in the phenylpropanoid, stilbenoid, and flavonoid pathways, as well as anthocyanin vacuolar transport and accumulation, are significantly upregulated by VviNAC17 and accumulate significantly more anthocyanins and proanthocyanidins in grapes [223]. VvNAC17 can activate *VvDREB1A* and *VvUFGT* expression by binding to their promoter to cause monomer anthocyanin to positively regulate drought tolerance in grapes [224]. MdNAC52 regulates proanthocyanidin biosynthesis by activating the transcription of *MdMYB9*, *MdMYB11*, and *MdLAR* (leucoanthocyanidin reductase) in apples [225]. Flavonoid accumulation also increases in *SlNAC12*-overexpressing tomatoes and enhances salt stress tolerance [212]. The MADS-box protein SlTAGL1 regulates a ripening-associated *SlDQD/SDH2* involved in flavonoid biosynthesis and resistance against *Botrytis cinerea* in post-harvest tomato fruit [226], but in *Ganoderma lucidum*, GlMADS1 negatively regulates flavonoid accumulation [227]. AcbHLH144 and SmbHLH60 negatively regulate phenolic biosynthesis in pineapples [228] and *S. miltiorrhiza* [229]. The MicroTom Metabolic Network (MMN) dataset was utilized to elucidate the dual roles of SlbHLH95 in regulating flavonoid metabolism and enhancing resistance to *Botrytis cinerea* in tomato fruits [213]. The heterologous expression of HlbZIP1A and HlbZIP2 promotes the accumulation of flavonol glycosides, PCs, and anthocyanins in *Petunia hybrida* [230]. WRKYs also participate in regulating various flavonoids. For instance, the expression of *VqWRKY56* activates genes involved in proanthocyanidin biosynthesis, including *VvCHS3*, *VvLAR1*, and *VvANR*, as reported by Wang et al. [231].

## 8. Discussion

With technological advancements, the integration of high-throughput omics technologies—including transcriptomics, proteomics, metabolomics, and epigenomics—has become indispensable in plant research, particularly for determining PC metabolism in horticultural crops [61,172,192,232]. These approaches enable the systematic profiling of gene expression levels, protein abundances, metabolite dynamics, and epigenetic modifications, facilitating the identification of key regulatory nodes and metabolic pathways underlying phenolic biosynthesis. However, current research still faces significant technical bottlenecks: single-omics analysis proves insufficient for unraveling complex regulatory networks, the integrated analysis of omics technologies lacks standardized data protocols, and high false-positive rates in small-sample studies restrict the precise construction of “gene–metabolite” regulatory networks. Specifically, the decoupling of transcriptomic and metabolomic datasets hinders the identification of causal relationships between gene expression and metabolic changes. The absence of unified preprocessing pipelines (e.g., normalization methods and statistical thresholds) across different studies leads to the poor comparability of combined analysis results.

In addition, modern biotechnology—including genetic engineering, cell engineering, and enzyme engineering—has provided innovative solutions for the regulation of PCs. Genetic engineering, such as agrobacterium transformation (e.g., *Rhizobium radiobacter* and *Rhizobium rhizogenes*) and gene editing (e.g., ZFN, TALEN, and CRISPR/Cas9), has been combined to reveal the molecular regulatory network mechanisms of PCs in horticultural plants and assist in breeding to improve their traits. Multiple studies have demonstrated that phenolic compound accumulation can be significantly enhanced through the use of *Rhizobium rhizogenes*-mediated hairy root cultures in a diverse range of plant species [233,234]. Furthermore, gene editing technology such as CRISPR/Cas9 has emerged as a transformative tool in regulating PC metabolism in horticultural crops, including anthocyanin [235], phenolic acid [229], stilbenes [236], lignins [237], and proanthocyanidins [238]. However, for transgenic technology, off-target effects, inefficient transformation of perennial crops, and the complexity of PC metabolic networks remain primary challenges. In regulatory contexts, ambiguous cross-border definitions, inadequate public awareness, and intellectual property disputes have posed bottlenecks to commercialization.

## 9. Conclusions and Future Directions

Future research on PCs in horticultural plants holds great potential and challenges. It is necessary to deeply analyze the molecular mechanisms regulated by PCs and develop efficient extraction, separation, and identification technologies; expand the research on their functions in plant growth, development, and stress adaptation, including fine growth regulation, multiple stress responses, and interactions with microorganisms and other secondary metabolites; and promote their application in the horticultural industry, facilitating quality improvement, post-harvest preservation, and ecological cultivation.

Looking forward, integrating multi-omics, high-fidelity editing technologies, and nano-delivery systems with synthetic biology to engineer PC biosynthetic pathways offers promise for crop improvement, thereby propelling the high-quality development of the horticultural industry.

## Figures and Tables

**Figure 1 ijms-26-05767-f001:**
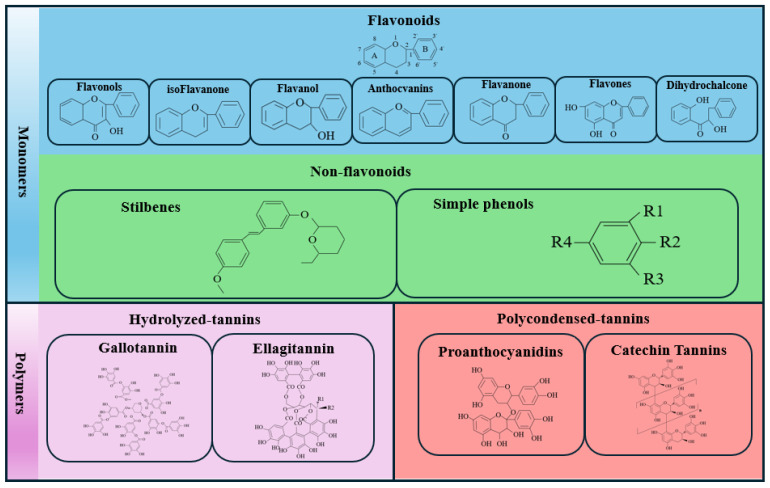
Classification of PCs and molecular structures of representative substances.

**Figure 2 ijms-26-05767-f002:**
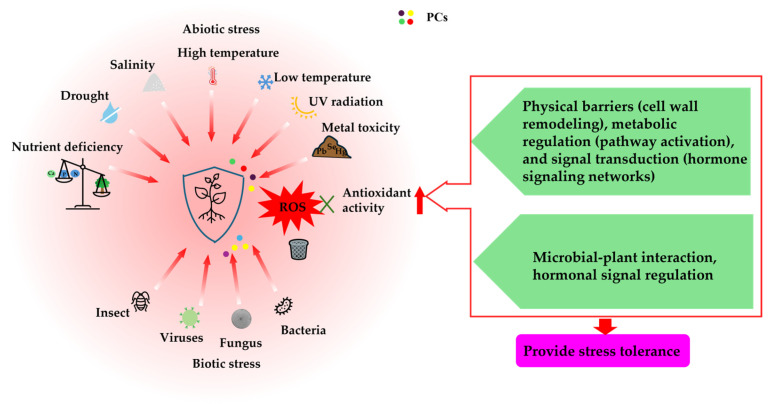
Schematic diagram of the action of phenolic compounds for stress resistance in plants. A single red arrow denotes enhancement, whereas a gradient red arrow signifies the generation of a stimulating effect.

**Table 1 ijms-26-05767-t001:** Main PCs in diverse horticultural plants.

Classification	Species	Content Range (g/Kg Fresh Weight)	References
Anthocyanin	Strawberry	0.15–0.60	[143]
	Goji	0.24–72.86	[144]
	Blueberry	1.00–2.00	[145]
	Grape	0.43–4.99	[146]
Flavone/Flavonol	Onion	0.10–13.59	[147]
	Mexican oregano	9.01–11.37	[148]
	Chamomile	3.00–5.00	[149]
	Peas	0.98–1.45	[147]
	Cranberry	1.49	[150]
Flavanone/Flavanonol	Citrus	0.10–6.30	[151]
	Tomato	1.3–22.2 × 10^−3^	[152]
Flavanols	Apple	0.032–1.66	[153]
Chalcone	Ashitaba	2.45–266.70 × 10^−3^	[154]
	safflower	55.00	[155]
Phenolic acid	Eggplant	3.20 × 10^−3^	[156]
	Carrot	2.95 × 10^−3^	[156]
Stilbenes	Grape	0.05–0.1	[157]
Tannin	Tea	6.00–14.00	[158]
	Persimmon	0.27–1.65	[159]

**Table 2 ijms-26-05767-t002:** Changes in PCs under various stressors in horticultural plants.

Classification	Stressors	Content Change (Fold/Increased)	Species	PCs	References
Abiotic stress	High temperature	1–1.80	Yarrow	Tannins	[177];
		0–0.65	Tomato	Flavonols	[104]
	Low temperature	-	Strawberry	Anthocyanin	[112];
		1.00–2.00	Apple		[178];
		42.39–158.31 mg/kg	Cabbage		[179]
	Salinity	≥2.00	Rose	Flavonoids	[177]
	Drought	0.30–2.88	Tomato	Flavonoids	[180]
	Metal toxicity	2.60–5.00	Cucumber	Epicatechin and flavone	[181]
	UV radiation	-	Grape	Flavonol	[104];
		-	Cucumber	phenolic acids	[182]
	Nutrient deficiency	1.00–3.00	Lettuce	Flavonoids and phenolic acids	[183]
Biotic stress	Aphids	1–2.82	Pepper	Cinnamic acids	[184]
	Viruses	1.26–1.58	Cucumber	Flavonoids	[185]
	Powdery Mildew	1–17.00	Grape	Stilbenes; flavonoids	[186]
	Bacteria	-	Apple	Phenolic acids	[187]

Red indicates content increase, green indicates content decrease, yellow indicates fluctuating levels (both increase and decrease), and “-” represents the absence of specific data.

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
