# Peer review of "A Comprehensive Review of Phenolic Compounds in Horticultural Plants"

_ijms, 2025, doi:10.3390/ijms26125767_

Round 1
Reviewer 1 Report
Comments and Suggestions for Authors
The review paper is focused on phenolic compounds’ (PCs) role in horticultural plant growth and physiology under adversity stress. This review provides a comprehensive synthesis of current knowledge on PCs, beginning with their detailed classification into major subgroups including flavonoids, simple phenols, stilbenes and polymers. In addition, metabolic pathways governing PCs biosynthesis, and their regulation at molecular and enzymatic levels, and evaluates the extraction and purification methodologies, from conventional solvent extraction to emerging technologies like ultrasound-assisted and supercritical fluid extraction. Furthermore, the complex regulatory networks controlling PCs metabolism are discussed, including transcription factors.
In my opinion, the manuscript is well-organized and presents an interesting and valuable review on the topic. However, during the minor revision stage, I suggest expanding and deepening the section related to the regulatory networks of phenolic compounds (PCs). Although the manuscript covers a broad scope, it would benefit from clarifying integrative tools such as omics technologies (e.g., genomics, transcriptomics, metabolomics), gene network analysis, or pathway modelling. The inclusion of such results would enhance the scientific depth and highlight the use of advanced, system-level approaches to understanding the regulation and metabolic dynamics of PCs.
Author Response
|
Comments 1: The review paper is focused on phenolic compounds’ (PCs) role in horticultural plant growth and physiology under adversity stress. This review provides a comprehensive synthesis of current knowledge on PCs, beginning with their detailed classification into major subgroups including flavonoids, simple phenols, stilbenes and polymers. In addition, metabolic pathways governing PCs biosynthesis, and their regulation at molecular and enzymatic levels, and evaluates the extraction and purification methodologies, from conventional solvent extraction to emerging technologies like ultrasound-assisted and supercritical fluid extraction. Furthermore, the complex regulatory networks controlling PCs metabolism are discussed, including transcription factors. In my opinion, the manuscript is well-organized and presents an interesting and valuable review on the topic. However, during the minor revision stage, I suggest expanding and deepening the section related to the regulatory networks of phenolic compounds (PCs). Although the manuscript covers a broad scope, it would benefit from clarifying integrative tools such as omics technologies (e.g., genomics, transcriptomics, metabolomics), gene network analysis, or pathway modelling. The inclusion of such results would enhance the scientific depth and highlight the use of advanced, system-level approaches to understanding the regulation and metabolic dynamics of PCs.
|
|
Response 1: Thank you for pointing this out. We agree with this comment. Therefore, we have added some references related to omics analyses of phenolic substances in horticultural plants to the descriptions of each section in the manuscript. Furthermore, a separate dedicated paragraph has been included to expand the Discussion, including content on the integration of modern omics technologies into research on horticultural phenolic substances. In fact, many research papers cited during the writing process involve omics technologies, without being reflected in the main text. The changes made to the revised manuscript can be found in Page 12 (line 474-477), Page 14 (line 555-556), Page 15 (line 595-598) and Page 15 (line 603-619), which are highlighted in red font. |

Reviewer 2 Report
Comments and Suggestions for Authors
This manuscript provides a thorough and well-structured review of phenolic compounds (PCs) in horticultural plants, integrating biochemical, physiological, and molecular insights. It offers a valuable interdisciplinary perspective on PC classification, biosynthesis, extraction methods, and their roles in response to both abiotic and biotic stressors. However, to meet the standards of a high-impact review article, several key areas require substantial improvement. These include the depth of synthesis, consistency in terminology, mechanistic explanations, and enhanced practical relevance through species-specific analysis and quantitative comparisons.
Recommendation:major revisions
- The manuscript offers a comprehensive and interdisciplinary overview of phenolic compounds (PCs) in horticultural plants, covering classification, biosynthesis, extraction methods, physiological roles, and molecular regulation under stress conditions.
• It effectively integrates perspectives from biochemistry, molecular biology, and agronomy.
• Both abiotic (e.g., drought, salinity, UV radiation, metal toxicity, nutrient deficiency) and biotic stresses (e.g., pathogens, insects) are well addressed, highlighting the versatile functions of PCs.
• The discussion includes transcription factors (e.g., MYB, NAC, ERF), signaling pathways, and gene regulatory networks, adding depth to the molecular understanding of PC biosynthesis.
• Inclusion of recent studies (2020–2024), especially those utilizing CRISPR/Cas9 and omics technologies, reflects up-to-date advances in plant secondary metabolism.
• The use of figures (e.g., PC classification, stress response pathways) and well-organized subheadings enhances clarity and readability.
• Sections such as "Regulatory Networks" and "Environmental Influences" structure complex content effectively.
• The manuscript follows a coherent and logical flow from basic classification to functional roles and future perspectives.
While you have to address the responses and implement improvements based on these comments.
- While the literature coverage is broad, there is minimal synthesis or meta-analysis. Add quantitative summaries (e.g., fold changes in PC accumulation, stress-response data) and tabular comparisons of species-specific responses under different stressors.
- Statements about temperature effects on PC accumulation (e.g., induction vs. inhibition) are inconsistent.Clarify whether differences arise from species- or tissue-specific responses. Provide context-dependent explanations.
- PCs are variably described as “secondary metabolites,” “antioxidants,” and “signaling molecules.” Define these terms clearly and use consistently, indicating the functional context in which each term applies.
- Claims such as “PCs universally enhance stress tolerance” oversimplify their roles. Avoid universal statements; discuss complex, dual roles (e.g., in defense and symbiosis), and provide mechanistic evidence where possible.
- Analysis is broad but lacks depth on key crops. Expand discussion on model or economically important species (e.g., tomato, grapevine, apple) for more practical relevance.
- CRISPR and RNAi are mentioned but not thoroughly explored. Discuss practical applications and current challenges in modulating PC traits via gene editing in horticultural crops.
- Overlap between sections on abiotic stress and environmental factors. Merge or streamline overlapping content to improve narrative flow and avoid repetition.
- Advanced methods (e.g., ultrasound, supercritical COâ‚‚) are introduced but lack performance data. Include comparative data on yield, cost-efficiency, and purity for each method.
- Several sections infer a direct correlation between antioxidant activity and stress tolerance without mechanistic detail. Support such claims with specific pathways or experimental findings.
- PCs are noted as both antimicrobial and symbiosis facilitators. Provide a mechanistic explanation for this functional duality.
Author Response
- Comments 1: While the literature coverage is broad, there is minimal synthesis or meta-analysis. Add quantitative summaries (e.g., fold changes in PC accumulation, stress-response data) and tabular comparisons of species-specific responses under different stressors.
Response 1: Thank you for pointing this out. We agree with this comment. Quantitative summaries (e.g., fold changes in PC accumulation, stress response data) and tabular comparisons of species-specific responses under different stressors have been supplemented with two tables, accordingly. The changes made to the revised manuscript can be found in Page 11 (line 450-451) and Page 12 (line 478-479), which are highlighted in red font. See also the tables below (Table 2 was not marked in red, since distinct colors in the table are employed to signify different content categories).
Table 1 Contents of main PCs in diverse horticultural plants
Table 2 Change of PCs under various stressors in horticultural plants
- Comments 2: Statements about temperature effects on PC accumulation (e.g., induction vs. inhibition) are inconsistent. Clarify whether differences arise from species- or tissue-specific responses. Provide context-dependent explanations.
Response 2: Thank you for pointing this out. We agree with this comment. Relative context-dependent explanations have been added to the manuscript, which can be found on Page 13, lines 528–534, where the revisions are highlighted in red font:
The differential accumulation of phenolic compounds in response to temperature reflects an adaptive strategy formed during the long-term evolution of plants, which is coordinately regulated by the ‘’species gene-tissue function-environmental signal-metabolic trade-off’’ framework.
- Comments 3: PCs are variably described as “secondary metabolites,” “antioxidants,” and “signaling molecules.” Define these terms clearly and use consistently, indicating the functional context in which each term applies.
Response 3: Thank you for pointing this out. We agree with this comment and have corrected the nomenclature relating to PCs throughout the manuscript in accordance with the suggestions.
As a class of plant secondary metabolites, PCs are generally named as “secondary metabolites” when describing their categories, metabolic pathways, contents and extraction methods. Meanwhile, during plants' responses to various stresses, PCs function as antioxidants, scavenging free radicals through hydrogen donation from phenolic hydroxyl groups to protect cells—a functional attribute denoted by "antioxidants" in relevant descriptions. Additionally, PCs act as signaling molecules, participating in intra- and intercellular signal transduction to regulate gene expression (e.g., salicylic acid). Their mechanisms operate via receptor recognition or metabolic pathway activation and, when conducting holistic or in-depth analyses of regulatory networks, their roles are typically defined as "signaling molecules." The above terminological definitions have been uniformly adjusted throughout the manuscript.
Comments 4: Claims such as “PCs universally enhance stress tolerance” oversimplify their roles. Avoid universal statements; discuss complex, dual roles (e.g., in defense and symbiosis), and provide mechanistic evidence where possible.
Response 4: Thank you for pointing this out. We agree with this comment. The manuscript now includes analyses of the physiological roles of phenolic substances in defense and symbiotic processes under biological stress, alongside explorations of the associated mechanistic studies. The modified part can be found on Page 9, lines 373–395, shown in red font.
- Comments 5: Analysis is broad but lacks depth on key crops. Expand discussion on model or economically important species (e.g., tomato, grapevine, apple) for more practical relevance.
Response 5: Thank you for pointing this out. We agree with this comment. The original manuscript predominantly discussed horticultural model plants, referencing 44 studies that employed apples, grapes, and tomatoes as research subjects. In accordance with your suggestions, the revised manuscript has incorporated 10 additional articles from top-tier journals published within recent years, along with an in-depth discussion (Guo et al., 2016; Radl et al., 2019; Yu et al., 2022b; Blancquaert et al., 2019; Cheng et al., 2023; Li et al., 2006; Lai et al., 2024; Šikuten et al., 2022; Yu et al., 2022a; Conti et al., 2022; Su et al., 2025; Vallverdú – Queralt et al., 2011).
- Comments 6: CRISPR and RNAi are mentioned but not thoroughly explored. Discuss practical applications and current challenges in modulating PC traits via gene editing in horticultural crops.
Response 6: Thank you for pointing this out. We agree with this comment. Relevant research progress regarding practical applications and current challenges in modulating PC traits via gene editing in horticultural crops has been supplemented in the Discussion (Page 15, lines 620–635, highlighted in red font):
Modern biotechnology (e.g., ZFN, TALEN, CRISPR/Cas9) should be combined to reveal the molecular regulatory network mechanism of PCs in horticultural plants and assist in breeding to improve the traits of PCs. Yet for transgenic technology, off-target effects, inefficient transformation of perennial crops, and the complexity of PCs metabolic networks remain primary challenges. In regulatory contexts, ambiguous cross-border definitions, inadequate public awareness, and intellectual property disputes have posed bottlenecks to commercialization. Looking forward, integrating multi-omics, high-fidelity editing technologies, and nano-delivery systems with synthetic biology to engineer PC biosynthetic pathways offers promise for crop improvement, thereby propelling the high-quality development of the horticultural industry.
- Comments 7: Overlap between sections on abiotic stress and environmental factors. Merge or streamline overlapping content to improve narrative flow and avoid repetition.
Response 7: Thank you for pointing this out. We agree with this comment. The second section of ‘environmental factors’ in Part 6 (Factors affecting the accumulation of PCs) has been renamed ‘External Factors’, in order to distinguish it from abiotic stress in the functional analysis section (Page 12, line 491). The description of PC functions in abiotic stress responses has been simplified. Additionally, we have incorporated content related to the functional section (abiotic stress) into this part, and added details regarding research progress in representative horticultural plant species, encompassing fruit trees and vegetables (Page 15, lines 496–512).
- Comments 8: Advanced methods (e.g., ultrasound, supercritical COâ‚‚) are introduced but lack performance data. Include comparative data on yield, cost-efficiency, and purity for each method.
Response 8: We appreciate your suggestion to include performance comparisons. In the revision, we have supplemented the text with some contents relating to yield, cost, and purity data for ultrasound, supercritical COâ‚‚, and control methods, supported by references to recent comparative studies (Shi et al., 2022; Tzani et al., 2023; Fu et al., 2021; Bezerra & Koblitz, 2025). The revised parts can be found on Page 4 (lines 163–167), Page 5 (lines 177–179; 205–208; 219–220), and Page 6 (lines 225–229), where the revisions are highlighted in red font.
- Comments 9: Several sections infer a direct correlation between antioxidant activity and stress tolerance without mechanistic detail. Support such claims with specific pathways or experimental findings.
Response 9: Thank you for pointing this out. We agree with this comment, and have supplemented the text with some content detailing the correlation between antioxidant activity and stress tolerance in the part ‘5.2 Antioxidant capacity’ (Page 8–9, lines 313–323 in red font):
Plants accumulated of various signal substances to adapt and survive in response to environmental cues and various kinds of abiotic stresses (temperature, drought, salinity, alkalinity, UV) (Figure 2), or pathogens/herbivores, which cause severe damage to the plants. ROS (reactive oxygen species) is an inevitable consequence of aerobic activities. Oxygen free radicals including superoxide (O2-), hydroxyl (OH·), and peroxyl (ROO·), as well as non-radical ROS including hydrogen peroxide (H2O2), singlet oxygen (1O2), and ozone (O3) (Hasanuzzaman et al., 2019). Under normal conditions, trace amounts of ROS can serve as signaling molecules for responding various stresses. Under severe stresses, due to ROS over-accumulation as a result of the dysregulation of ROS maintaining mechanisms, ROS can directly attack biological macromolecules, and leading to oxidative damage to plasma membranes, nucleic acids, and proteins, even if directs to the oxidative obliteration of the cell (Hasanuzzaman et al., 2019; Mittler, 2002). Enhancing plant antioxidant capacity helps improve plant stress resistance.
- Comments 10: PCs are noted as both antimicrobial and symbiosis facilitators. Provide a mechanistic explanation for this functional duality.
Response 10: We appreciate your feedback on this point. The manuscript now includes a mechanistic explanation regarding the duality of antimicrobial and symbiosis-facilitating functions, supplemented on Page 9 (lines 373–395) and highlighted in red font:
On the one hand, root-secreted PCs exhibit dual functionality: inhibiting detrimental microorganisms while enriching and stabilizing beneficial symbiotic microbial communities. As for antibacterial properties, it can disrupt the cell membranes of microorganisms and interfere with their metabolism. Furthermore, PCs also reinforcing the cell wall at the challenge site, and also accompanied by ROS driving cell wall cross linking, antimicrobial activity and defense signaling (Field et al., 2006; Mandal et al., 2010) Multiple simple and complex PCs, including cajanin, medicarpin, glyceolin, rotenone, coumestrol, phaseolin, phaseolinin, flavonoids, act as phytoalexins, phytoanticipins and nematicide (Ndakidemi et al., 2003; Mandal et al., 2010). The polyphenol extract from flaxseed exhibits antibacterial effects against Staphylococcus aureus, Escherichia coli, Listeria monocytogenes, Salmonella and Pseudomonas fluorescens. Its antibacterial mechanism mainly involves disrupting the cell membranes and cell walls of microorganisms, thus interfering with their normal life activities (Wang et al, 2024). In addition, microorganisms directly or indirectly affected the quality and quantity of rhizosphere localized PCs through modification the exudation patterns and microbial catabolism, and influencing the plant-microbe interaction (Shaw et al., 2006; Mandal et al., 2010). In terms of specific symbiotic relationships, the PCs secreted by plant roots can alter the rhizosphere microbial community, promoting the growth of beneficial bacteria and inhibiting pathogenic bacteria. PCs esprcially flavonoids, secreted from symbiotic legumes, can serve as chemoattractants for guiding the growth of rhizobial cells (Mandal et al., 2010). Acting as the signaling molecular, host root secretes PCs can regulated nod gene expression by the symbiont (Rhizobium) and so modify the legume-rhizobial symbiosis (Mandal et al., 2010). Phenolic acids such as p-coumaric, caffeic, and ferulic acidsserve as novel form of nod gene inducer in the L. japonicus-Mesorhizobium symbiosis (Shimamura et al, 2022).
- Response to Comments on the Quality of English Language
Response: Thank you for your comments on the English language quality. We have thoroughly revised the manuscript to enhance its linguistic precision and readability.
Certification of English Language Editing:

Reviewer 3 Report
Comments and Suggestions for Authors
This manuscript introduces the classification, biosynthesis, regulation, extraction, and purification of phenolic compounds (PCs), and further compiles the roles of PCs in plant stress responses. However, the author's topic is too broad, resulting in a manuscript that lacks focus and fails to establish relevance to Horticultural Plant. The author should purposefully summarize and discuss key issues in practical production or research areas of horticultural plants.
Furthermore, a review should not merely list existing research findings; it must synthesize core conclusions, identify existing problems, and propose feasible solutions or research directions based on the current research landscape.
Additionally, parts of the abstract are verbose and lack conciseness. For example: "This review provides a comprehensive synthesis of current knowledge on PCs, beginning with their detailed classification into major subgroups including flavonoids, simple phenols, stilbenes, and phenolic polymers (e.g., lignins, tannins)." Is this directly relevant to the manuscript’s core theme?
Other Issues:
(1) The author should carefully review linguistic expressions. For instance: "Solvent extraction, capitalizes on the varying solubility of PCs in different solvents, making it the most traditional and widely adopted extraction method. (Lines 158-189)"
Setting aside grammatical issues, what logical connection exists between "Solvent extraction..." and "the most traditional and widely adopted extraction method"? This is clearly illogical.
(2) The manuscript’s "Figure 2. Schematic diagram of the phenolic compound against stress resistance in plants" provides very limited information. The author should summarize key relevant insights and enrich the schematic’s content.
Author Response
Response to Reviewer 3 Comments
|
1. Summary |
|
|
|
||
|
Thank you very much for taking the time to review this manuscript. Please find our detailed responses below and the corresponding revisions as tracked changes in the re-submitted files. |
|
||||
|
2. Questions for General Evaluation |
Reviewer’s Evaluation |
Response and Revisions |
|||
|
Is the work a significant contribution to the field? |
|
||||
|
Is the work well organized and comprehensively described? |
|
||||
|
Is the work scientifically sound and not misleading? |
|
||||
|
Are there appropriate and adequate references to related and previous work? |
|
||||
|
Is the English used correct and readable? |
The English is fine and does not require any improvement.
|
|
|||
|
3. Point-by-point response to Comments and Suggestions for Authors |
|
||||
Comments and Suggestions for Authors:
This manuscript introduces the classification, biosynthesis, regulation, extraction, and purification of phenolic compounds (PCs), and further compiles the roles of PCs in plant stress responses. However, the author's topic is too broad, resulting in a manuscript that lacks focus and fails to establish relevance to Horticultural Plant. The author should purposefully summarize and discuss key issues in practical production or research areas of horticultural plants.
Furthermore, a review should not merely list existing research findings; it must synthesize core conclusions, identify existing problems, and propose feasible solutions or research directions based on the current research landscape.
Additionally, parts of the abstract are verbose and lack conciseness. For example: "This review provides a comprehensive synthesis of current knowledge on PCs, beginning with their detailed classification into major subgroups including flavonoids, simple phenols, stilbenes, and phenolic polymers (e.g., lignins, tannins)." Is this directly relevant to the manuscript’s core theme?
Other Issues:
(1) The author should carefully review linguistic expressions. For instance: "Solvent extraction, capitalizes on the varying solubility of PCs in different solvents, making it the most traditional and widely adopted extraction method. (Lines 158-189)"
Setting aside grammatical issues, what logical connection exists between "Solvent extraction..." and "the most traditional and widely adopted extraction method"? This is clearly illogical.
(2) The manuscript’s "Figure 2. Schematic diagram of the phenolic compound against stress resistance in plants" provides very limited information. The author should summarize key relevant insights and enrich the schematic’s content.
- Comments 1: However, the author's topic is too broad, resulting in a manuscript that lacks focus and fails to establish relevance to Horticultural Plant. The author should purposefully summarize and discuss key issues in practical production or research areas of horticultural plants.
Response 1: Thank you for your professional suggestions. We agree with this comment. When drafting the manuscript, we intended to focus on the functional aspects of PCs in horticultural plants. However, during the literature review, we identified several issues, including confusion in the nomenclature and classification of PCs. Therefore, we believe that it is essential to systematically elaborate on the main classifications, content distribution characteristics, extraction methods, functions, and regulatory networks of phenolic compounds. Nevertheless, this approach led to a broad scope of content, potentially causing a lack of clear focus.
Despite this, we maintain that such a review is of significant importance for the study of PCs in horticultural disciplines, as well as for basic research and industrial applications in this field. Consequently, we have revised the title to "A Comprehensive Review of Phenolic Compounds in Horticultural Plants". Additionally, we have inserted two tables: one providing a quantitative summary of the content ranges of PCs in major horticultural crops, and the other focusing on species-specific stress responses under different stressors. The changes in the revised manuscript can be found on Page 11 (lines 450–451) and Page 12 (lines 478–479), which are highlighted in red font. Moreover, we have added two separate sections to the original manuscript, in which we summarize and discuss key issues in practical production and research areas of horticultural plants, which can be found on Page 15, lines 623–638, and highlighted in red font.
- Comments 2: Furthermore, a review should not merely list existing research findings; it must synthesize core conclusions, identify existing problems, and propose feasible solutions or research directions based on the current research landscape.
Response 2: Thank you for your insightful comments, which we fully endorse. The original manuscript indeed failed to thoroughly address the current challenges relating to PCs in horticultural plants, nor did it elaborate on the key developmental directions for future research and applications in this context. In response, we have added two dedicated sections to the revised manuscript to summarize and discuss critical issues in both practical production and research domains of horticultural plants. These additions are located on Page 15, Lines 603–635, and are highlighted in red for clarity.
- Comments 3: Additionally, parts of the abstract are verbose and lack conciseness. For example: "This review provides a comprehensive synthesis of current knowledge on PCs, beginning with their detailed classification into major subgroups including flavonoids, simple phenols, stilbenes, and phenolic polymers (e.g., lignins, tannins)." Is this directly relevant to the manuscript’s core theme?
Response 3: Thank you for pointing this out. We agree with this comment. We have restructured the language to enhance and streamline the abstract, as shown below:
Phenolic compounds (PCs), key secondary metabolites in horticultural plants, are structurally categorized into flavonoids, simple phenols, stilbenes, and tannins. Synthesized via the shikimate and phenylpropanoid pathways, PCs are regulated in their metabolism by transcription factors (e.g., MYB, bZIP) and influenced by genetic backgrounds and environmental stresses (e.g., temperature, UV), thereby leading to species- or tissue-specific distribution patterns. Advanced extraction/separation techniques (e.g., ultrasonic-assisted, HPLC) have enabled systematic PC characterization. Functionally, PCs enhance plant stress resistance (abiotic/biotic) through antioxidant activity, cell wall reinforcement, and defense signaling. Their dual roles as ROS scavengers and signaling molecules are highlighted. This review synthesizes PC classification, metabolic regulation, and biological functions. It provides a scientific basis for improving PC content in horticultural plants, aiming to enhance stress resilience, postharvest and storage quality, and nutritional value for sustainable agriculture.
- Comments 4: The author should carefully review linguistic expressions. For instance: "Solvent extraction, capitalizes on the varying solubility of PCs in different solvents, making it the most traditional and widely adopted extraction method. (Lines 158-189)" Setting aside grammatical issues, what logical connection exists between "Solvent extraction..." and "the most traditional and widely adopted extraction method"? This is clearly illogical.
Response 4: Thank you for pointing this out. We agree with this comment. We sincerely apologize for the language issues in our manuscript, which resulted from our negligence and inexperience. However, we have thoroughly reviewed and corrected the entire paper, and have even invited professional language instructors to assist in its revision. Regarding the specific expression error, we intended to clarify that solvent-based extraction (relying on the solubility of phenolic compounds in different solvents) is the simplest and most fundamental method, and all modern extraction techniques are essentially optimized versions of this approach, often combined with auxiliary methods like ultrasonication. We have made the following corrections in the paper: ‘Solvent extraction, leveraging the solubility behavior of phytochemicals, remains the simplest and most foundational technique, with all modern methods essentially building upon its principles’ (Page 4, Lines 153-155).
- Comments 5: The manuscript’s "Figure 2. Schematic diagram of the phenolic compound against stress resistance in plants" provides very limited information. The author should summarize key relevant insights and enrich the schematic’s content.
Response 5: We sincerely appreciate your professional suggestions, which we believe will significantly enhance the persuasiveness of our review. In conjunction with the revised manuscript content, we have enhanced and refined the figures, as displayed below.

Round 2
Reviewer 2 Report
Comments and Suggestions for Authors
The revised manuscript shows noticeable improvement in structure and clarity. The title is now more coherent and better aligned with the content. The discussion has been strengthened, and the overall presentation is clearer.
Overall, the manuscript is acceptable for publication in its current form.
Reviewer 3 Report
Comments and Suggestions for Authors
The author's revised manuscript has resolved my concerns. I have no other questions.